# The Role of Copper Intake in Bone Health: A Quantitative Analysis in Postmenopausal Spanish Women

**DOI:** 10.3390/ejihpe15020025

**Published:** 2025-02-19

**Authors:** María Luz Canal-Macías, Luis Manuel Puerto-Parejo, Jesús María Lavado-García, Raúl Roncero-Martín, Juan Diego Pedrera-Zamorano, Fidel López-Espuela, Purificación Rey-Sánchez, Antonio Sánchez-Fernández, José M. Morán

**Affiliations:** 1Metabolic Bone Diseases Research Group, Nursing Department, Nursing and Occupational Therapy College, University of Extremadura, 10003 Cceres, Spain; luzcanal@unex.es (M.L.C.-M.); luis.puerto@salud-juntaex.es (L.M.P.-P.); jmlavado@unex.es (J.M.L.-G.); rronmar@unex.es (R.R.-M.); jpedrera@unex.es (J.D.P.-Z.); fidellopez@unex.es (F.L.-E.); prey@unex.es (P.R.-S.); 2Servicio de Tocoginecologia, Hospital San Pedro de Alcántara, 10003 Caceres, Spain; gineantonio@gmail.com

**Keywords:** copper, osteoporosis, bone mineral density, dietary intake, food frequency questionnaire

## Abstract

(1) Background: Copper is a crucial trace element which is vital to growth and development and is especially important in bone health. Copper intake is now the focus of much broader research beyond its associations with nail growth, looking at copper’s potential in contributing to bone integrity to prevent a high risk of osteoporosis as well. (2) Methods: This study included postmenopausal women from a larger longitudinal study conducted between 2019 and 2022. Bone health was assessed using three quantitative techniques: heel QUS, DXA and pQCT. Copper intake was evaluated using a 131-item, 7-day food frequency questionnaire. Data from these assessments were used to analyze the relationship between copper intake and bone health. (3) Results: In the unadjusted multiple linear regression model, associations were found between copper intake levels and both BUA (dB/MHz) and pQCT cortical + subcortical density (mg/cm^3^), with copper intake acting as a negative predictor in both instances. However, these associations lost statistical significance after adjusting for participant age and weight. No further associations were identified for the other parameters assessed. (4) We conclude that our study does not reveal an association between copper intake and bone health in postmenopausal Spanish women.

## 1. Introduction

Osteoporosis is a systemic, metabolic skeletal disease characterized by markedly deficient bone mineral density (BMD) and excessive susceptibility to fracture. It is characterized by an imbalance of bone remodeling dynamics, with bone resorption outpacing bone formation to cause the collapse of structural integrity and loss of mechanical resilience in the skeletal framework ([23]). The estimated prevalence of osteoporosis in women based on total hip BMD varies from 9% in the UK to 15% in France and Germany and up to 16–38% when spine BMD is taken into account. The rates in men based on the hip varied from 1% for the UK to 4% for Japan and increased to 3–8% when spine BMD data were included ([38]). More recently, the global prevalence of osteoporosis was estimated to be 18.3% overall, 23.1% in women and 11.7% in men, with the highest regional prevalence recorded in Africa at 39.5% ([33]). Osteoporosis imposes a heavy financial burden on the health system, with high therapy costs and decreased productivity due to disease-related impairments. Direct medical costs, such as hospital admission, pharmacological factors and/or treatment and rehabilitative care, are considered within the overall financial impact; however, their neglect results in indirect costs, such as lost productivity time and nonproductivity time, including absenteeism and disability ([21]).

Various factors, such as a low body mass index (BMI), female sex, advanced age and heredity, are risk factors for the onset of osteoporosis and diminished BMD ([11]; [41]). Adjusting for age and weight is essential in studies linking dietary intake to bone health, as both are major determinants of BMD. Age-related bone loss, driven by decreased bone formation and hormonal changes, significantly impacts BMD ([9]). Similarly, a higher body weight is associated with greater BMD due to increased mechanical load, while a low weight elevates osteoporosis risk ([3]; [40]). Without accounting for these factors, regression analyses may produce biased associations. Accurate adjustments ensure that dietary influences on bone health are correctly interpreted. In addition to the well-established risk factors, nutritional ([14]), trace element and vitamin deficiencies in the development of osteoporosis are receiving increasing emphasis ([7]; [13]; [18]; [27]).

Copper is a trace element of vital importance for sustaining human health ([34]). Copper is necessary to maintain healthy growth and development and helps nourish bones, the brain, cardiovascular function and other vital organs. The human body cannot make copper, so we must rely entirely on the diet for this essential trace mineral ([8]; [12]; [17]; [35]; [37]). There have been few observational studies regarding the association between copper and osteoporosis risk, as previous studies have largely examined the association between serum copper and osteoporosis ([4]; [20]; [28]). However, recent studies have investigated the relationship between dietary copper intake and bone health in humans and attempted to determine how copper may be involved in safeguarding bone integrity and reducing osteoporosis risk ([6]; [8]; [12]; [26]). Copper is an important constituent of several physiological processes, the most important being the incorporation of copper into enzymes essential for energy metabolism and the formation of connective tissue crosslinks, particularly in bone ([32]). Copper supports osteogenesis by promoting the differentiation of bone mesenchymal stem cells toward bone formation rather than adipogenesis ([29]). In addition, copper deficiency can result in Menkes disease, with osteoporosis being one of its primary adverse effects ([5]; [6]; [25]).

In nutrition-based studies, dietary copper intake is usually estimated using dietary questionnaires. These types of questionnaires are specifically used to ask for detailed information about consumption habits and then analyze nutrient content with the intake of these foods, including copper. As an example, food frequency questionnaires were used to quantify dietary copper intake in more than 10,000 participants in the Atherosclerosis Risk in Communities study and were associated with cognitive outcomes in this sample ([39]). Questionnaires of dietary habits have been useful in uncovering the role of copper in diverse aspects of health. The relationship between dietary copper and cardiovascular health has become a topic of research; some studies show that both deficient and excessive copper levels may influence cardiovascular disease risk ([17]). Dietary questionnaires ([12]; [26]) have also been used to examine copper involvement in bone health. In the present study, we aimed to add to the body of knowledge about the association between dietary copper intake and bone health. Bone health determinations were performed using different complementary quantitative techniques, including dual-energy X-ray absorptiometry (DXA), peripheral quantitative computed tomography (pQCT) and calcaneal quantitative ultrasound (QUS). Using this approach, not only accurate BMD determinations but also the microarchitectural and mechanical characterization of bone was obtained, thereby allowing for a more specific assessment of the possible association of copper intake and bone health in postmenopausal women.

## 2. Materials and Methods

### 2.1. Participants and Sample Characteristics

This study takes place in the context of a larger longitudinal study between 2019 and 2022, during which the authors followed up with the participants. The data presented in the present study are from the cross-sectional analysis of the main study’s baseline measurements. The present study included 313 postmenopausal women. The participants in this study were community-dwelling women of white European descent who were not diagnosed with functional mental or physical disability as confirmed by their primary care physician or a chronic medical specialist in their care team. These patients did not require medications known to interfere with mineral metabolism (such as oral anticoagulants, antipsychotics or corticosteroids), and they did not suffer from diabetes mellitus; liver disease; renal osteodystrophy; associated disorders of mineral metabolism; or diseases of the parathyroid, thyroid, adrenal or ovarian glands. All participants provided written informed consent. The Ethical Advisory Committee of the University of Extremadura endorsed this study. All the participants provided written informed consent in accordance with the Declaration of Helsinki.

### 2.2. Heel Quantitative Ultrasound (QUS) Assessment

Quantitative ultrasound (QUS) was conducted using the Sahara Clinical Sonometer (Hologic, Bedford, MA, USA) following a standardized protocol. Trained staff ensured that the participants could complete measurements on both heels, excluding those with open wounds, injuries or metal implants in the heel. Daily quality control was performed using a phantom, as per the manufacturer’s guidelines. The device measured the speed of sound (SOS) and broadband ultrasound attenuation (BUA), indicators of bone health, with higher values indicating better bone quality ([36]).

### 2.3. Dual-Energy X-Ray Absorptiometry (DXA) Assessment

BMD at the femoral neck (FN), femoral trochanter (FT) and L2–L4, as well as at a combined L2, L3 and L4 region, was measured via dual-energy X-ray absorptiometry (DXA). Body weight and body height were also recorded, and body mass index (BMI) was also calculated. BMD values were assessed via densitometry with a NORLAND XR-800 device (Norland Medical Systems Inc., Fort Atkinson, WI, USA). The BMD values are expressed in grams per square centimeter ([1]; [30]).

### 2.4. Peripheral Quantitative Computed Tomography (pQCT) Assessment

Peripheral quantitative computed tomography (pQCT) scans of the nondominant distal forearm were obtained using a Stratec XCT-2000 scanner (Stratec Medizintechnik, Pforzheim, Germany). At 4% of the total forearm length, an image was taken with the scanner positioned at the distal end of the forearm. The data from the XCT-2000 scans were processed using the software package (version 5.50) provided by the manufacturer. The pQCT scans provide a volumetric measurement of bone mineral density and allow differentiation between trabecular and cortical bone ([30]).

### 2.5. Assessment of Copper Intake

Total dietary copper, vitamin D, calcium and energy intake were assessed via validated frequency questionnaires, as previously described ([16]; [31]). The participants completed a comprehensive 131-item, 7-day food frequency questionnaire. Food intake was quantified using a dietary scale, as well as measuring cups and spoons. The questionnaire was self-administered, with a response rate of 100%. Nutrient and energy intake values were assessed according to the Spanish food composition database ([22]).

### 2.6. Statistical Analysis

The median and interquartile range (IQR) were used to describe quantitative variables. For comparisons between two groups, the Mann–Whitney U test was used, and for comparisons between more than two groups, the Kruskal–Wallis test was used. The chi-square test was used to analyze the dependences of categorical variables. For some comparisons, participants were grouped on the basis of their DXA T score into low bone mass (T score < −1) or normal categories. Multiple linear regression analyses (Enter method) were performed using two models: one that was unadjusted on the basis of only copper intake quartiles and one that was adjusted for participant weight and age. Statistical significance was set at a *p* value < 0.05. All analyses were conducted using JASP software ([15]).

## 3. Results

Table 1 displays the characteristics of the study participants, categorized according to their bone health status of normal, osteopenic or osteoporotic. The prevalence of osteoporosis in the studied sample was 23%. Compared with participants without osteoporosis, those with osteoporosis were generally older and had lower BMIs, as well as smaller waist and hip measurements. Additionally, they had more years since menopause. No significant differences were observed between the groups regarding the number of pregnancies, smoking habits or dietary intake of vitamin D, calcium, energy or copper (Table 1).

To further explore this analysis, participants were subsequently grouped into low (T score < −1) and normal bone mass categories. Comparative analysis between groups with low bone mass and those with normal bone mass revealed significant differences in physical and age-related characteristics. Participants with lower bone mass presented significantly different values for body mass index, height, waist circumference, hip circumference and age. Conversely, no statistically significant differences were observed in reproductive variables or in the dietary intake of specific nutrients, including vitamin D, calcium or copper, or caloric intake (Table 2).

To further investigate the potential role of copper intake in bone health, participants were divided into quartiles on the basis of their copper intake levels. The results of the quantitative analysis of bone health using QUS, DXA and pQCT, stratified by copper intake quartiles, are presented in Table 3. No significant differences were observed in the quantitative bone health measurements among the copper dietary intake groups.

In the unadjusted multiple linear regression model, associations were observed exclusively between copper intake levels and both BUA (dB/MHz) and pQCT cortical + subcortical density (mg/cm^3^), with copper intake serving as a negative predictor in both associations (Table 4). However, these associations were not significant after the models were adjusted for participant age and weight. No additional associations were detected for the remaining parameters evaluated.

## 4. Discussion

In the present study, we explored the relationship between dietary copper intake and bone health in postmenopausal Spanish women via a comprehensive array of quantitative densitometric tests. No associations were observed between copper intake and the various bone health measurements included in this study. To our knowledge, this is the first study to evaluate the relationship between dietary copper intake and bone health using a battery of densitometric measures. Few epidemiological studies have examined serum copper levels and their relationship with bone health. Both lower and higher serum copper levels are associated with lower BMD (by %) in the total femur and femoral neck and an increased fracture risk, particularly in men ([28]).

A total of 728 postmenopausal women reporting a history of osteoporosis were investigated for relationships between the serum levels of nine important minerals and osteoporosis. Serum copper levels were significantly related to the diminished BMD of the total femur, femoral neck and lumbar spine, which suggests that mineral deficiency is a risk factor for osteoporosis in postmenopausal women ([24]). The serum copper levels did not significantly differ among the healthy, osteopenic and osteoporotic groups of 107 postmenopausal women. Additionally, serum copper levels were not related to BMD, and copper did not contribute directly to or hinder the bone health of these postmenopausal women ([2]).

A recent study reported negative findings regarding the effects of serum copper levels on bone health in younger populations, highlighting the need for further research to clarify these associations ([19]). Therefore, previous studies have investigated the extent to which serum copper correlates with BMD but not the extent to which dietary copper intake correlates with the development of osteoporosis or low bone mass.

Recently, observational studies evaluating the relationship between dietary copper intake and bone health as assessed by DXA measurements have been published. A study investigating the relationship between copper intake and bone health analyzed data from 8224 adults in the United States. A higher copper intake was associated with increased BMD at the femur and spine and a reduced risk of osteoporosis. Participants in the highest copper intake quartile had a 59% lower risk of osteoporosis than did those in the lowest quartile, indicating a positive role of dietary copper intake in bone health ([12]). A second study explored the association of BMD with copper and selenium intake in 522 women, 20 to 88 years of age. Lower BMD at various skeletal sites was associated with low intakes of both trace elements. After potential confounders were adjusted for, a low copper intake was linked to a 1.8–4.0% reduction in BMD, supporting the role of copper in maintaining optimal bone health ([26]). In this context, a controlled trial investigated the effects of copper supplementation on vertebral trabecular bone mineral density (VTBMD) over two years in 73 healthy women aged 45–56 years. The participants were randomly assigned to receive 3 mg of copper or a placebo. Although no significant effects on copper status biomarkers were observed, the copper-supplemented group showed no change in VTBMD, whereas the placebo group experienced a significant reduction in VTBMD ([10]). Other studies have reported less favorable findings as regards the role of copper and other minerals in bone health. In a study of postmenopausal women with osteoporosis and osteopenia, it was discovered that the dietary magnesium, zinc, calcium and copper intake was lower than the recommended levels. Nevertheless, no difference was found in the dietary intake of copper between groups with osteopenia and osteoporosis ([20]). Importantly, however, the results of this study are in agreement with those of the current study while illustrating a potential statistical power concern. The sample sizes included in studies recently published by [12] ([12]) and [26] ([26]) are noteworthy, and it cannot be ruled out that the absence of statistically significant findings in our study may be related to a possible type II error. Our findings here, suggesting that no associations were observed, could be due to insufficient power to detect subtle effects in our sample. The study by Pasco et al. 2024 reported a post hoc computed effect size strikingly similar to that reported in our own research. Secondly, they also used tertiles, which allowed for increased statistical power through increasing the sample size in separate groups so that the analysis was more sensitive in detecting conceivable effects.

It is evident from the analysis of the scientific literature currently available (mainly from observational studies) that there is a discrepancy in studies regarding how copper intake is associated with bone health. This divergence in results is based on the observational nature of most of the studies, the populations and the methodologies of analysis used in each case. We, therefore, consider it a priority that the next studies to be carried out approach the subject from a longitudinal point of view, assessing BMD and the risk of fracture in the context of copper intake in the medium and long term. We hope that these studies can help to establish without a doubt the existence of causality and try to understand the mechanisms by which copper participates in the regulation of bone metabolism. In addition, a systematic review of the available literature could help to clarify the relationship. These approaches could help to clarify in the future the need to improve or not improve dietary recommendations regarding copper intake or to discover possible therapeutic interventions that favor adequate copper intake, which is related to greater benefits in bone health in postmenopausal women.

We acknowledge that our study has several limitations inherent to observational research. One of the primary limitations is the use of a dietary recall questionnaire to assess copper intake, which is subject to recall bias and may not accurately capture long-term dietary habits. Additionally, the sample size in our study was relatively small, which poses a recognized risk of type II error, potentially limiting the ability to detect statistically significant associations. Despite these limitations, our study has notable strengths. This study is the first to incorporate data from three distinct quantitative techniques for assessing bone health in postmenopausal Spanish women, offering a comprehensive evaluation of bone mineral density across different methodologies. The use of three different quantitative techniques, QUS, DXA and pQCT, improves the completeness of the study. This is achieved through a multidimensional assessment of bone health in the women studied. Each technique individually determines a different characteristic, and together they represent a complementary approach to bone health. On the one hand, QUS offers an accessible, portable, non-ionizing option to assess the mechanical properties of bone tissue and fracture risk. On the other hand, DXA is recognized as the gold standard for the diagnosis of osteoporosis worldwide, providing accurate BMD determinations in those skeletal regions that are critical for diagnosis (lumbar spine and hip). Finally, pQCT analysis provides accurate information on bone microarchitecture, trabecular and cortical bone compartments are analyzed, and information on bone density and volume is acquired.

## 5. Conclusions

We conclude that our study does not reveal an association between copper intake and bone health in postmenopausal Spanish women. In light of recent findings from other studies, our results likely reflect a situation of possible type II error, attributed to insufficient statistical power. We recommend that these data be utilized in future studies, particularly within the context of meta-analyses, to further elucidate the potential role of copper intake in bone health in postmenopausal women.

## Figures and Tables

**Table 1 ejihpe-15-00025-t001:** Anthropometric, biological, dietary and lifestyle characteristics of the study sample according to the WHO osteoporosis classification.

Variable	Bone Health	n	Median/n	IQR/Percentage	*p* Value
BMI (kg/m^2^)	NORMAL	100	27.6	5.6	<0.001
	OSTEOPENIA	141	26.6	5.3	
	OSTEOPOROSIS	72	23.4	4.0	
Height (m)	NORMAL	100	1.59	0.08	0.088
	OSTEOPENIA	141	1.58	0.07	
	OSTEOPOROSIS	72	1.57	0.07	
Waist (cm)	NORMAL	100	91	16.5	<0.001
	OSTEOPENIA	141	87	13	
	OSTEOPOROSIS	72	81	11	
Hip (cm)	NORMAL	100	107	13	<0.001
	OSTEOPENIA	141	104	10	
	OSTEOPOROSIS	72	99	11	
Age (years)	NORMAL	100	58	6.5	0.019
	OSTEOPENIA	141	60	6	
	OSTEOPOROSIS	72	60	6.25	
Age at menarche (years)	NORMAL	100	12	3	0.917
	OSTEOPENIA	141	13	1	
	OSTEOPOROSIS	72	13	2	
Years since menopause (years)	NORMAL	100	7	10.25	<0.001
	OSTEOPENIA	141	10	7	
	OSTEOPOROSIS	72	11	8.25	
Pregnancies (n)	NORMAL	100	2	1	0.306
	OSTEOPENIA	141	2	1	
	OSTEOPOROSIS	72	2	1.25	
Number of children (n)	NORMAL	100	2	0	0.521
	OSTEOPENIA	141	2	1	
	OSTEOPOROSIS	72	2	1	
Smoker (Y/N) *	NORMAL	100	84/16	84/16	0.174
	OSTEOPENIA	141	111/30	78.7/21.3	
	OSTEOPOROSIS	72	52/20	72.2/27.8	
Vitamin D intake (IU/day) (Reference: 200 UI/day)	NORMAL	100	280	360	0.775
	OSTEOPENIA	141	280	320	
	OSTEOPOROSIS	72	280	520	
Calcium intake (mg/day) (Reference: 800 mg/day)	NORMAL	100	932	537	0.818
	OSTEOPENIA	141	973	651	
	OSTEOPOROSIS	72	934.5	804.5	
Energy (Kcal/day)	NORMAL	100	2026.8	891.7	0.432
	OSTEOPENIA	141	2042.9	926.5	
	OSTEOPOROSIS	72	2227.8	945.5	
Copper intake (mg/day) (Reference 1.3 mg/day)	NORMAL	100	1.1	2.1	0.789
	OSTEOPENIA	141	1.3	3.1	
	OSTEOPOROSIS	72	1.3	2.1	

Comparisons were performed using the nonparametric Kruskal-Wallis test. * The chi-square test was used for the variable “smoker”.

**Table 2 ejihpe-15-00025-t002:** Anthropometric, biological, dietary and lifestyle characteristics of the study sample with low or normal BMD.

Variable	Bone Health	n	Median/n	IQR/Percentage	*p* Value
BMI (kg/m^2^)	LOW	212	25.7	5.5	<0.001
	NORMAL	101	27.6	5.5	
Height (m)	LOW	212	1.58	0.07	0.039
	NORMAL	101	1.59	0.08	
Waist (cm)	LOW	212	85	13.3	<0.001
	NORMAL	101	91	16	
Hip (cm)	LOW	212	103	12	<0.001
	NORMAL	101	107	13	
Age (years)	LOW	212	60	6	0.003
	NORMAL	101	58	7	
Age at menarche (years)	LOW	212	13	2	0.769
	NORMAL	101	12	3	
Years since menopause (years)	LOW	212	10	7.3	0.822
	NORMAL	101	7	10	
Pregnancies (n)	LOW	212	2	1	0.093
	NORMAL	101	2	1	
Number of children (n)	LOW	212	2	1	0.189
	NORMAL	101	2	0	
Smoker (Y/N) *	LOW	212	162/50	76.4/23.6	0.116
	NORMAL	101	85/16	84.2/15.8	
Vitamin D intake (IU/day) (Reference: 200 UI/day)	LOW	212	280	360	0.72
	NORMAL	101	280	400	
Calcium intake (mg/day) (Reference: 800 mg/day)	LOW	212	968	692.3	0.53
	NORMAL	101	934	535	
Energy (Kcal/day)	LOW	212	2156	943.6	0.344
	NORMAL	101	2007.4	898.7	
Copper intake (mg/day) (Reference: 1.3 mg/day)	LOW	212	1.3	2.8	0.57
	NORMAL	101	1.2	2.2	

The comparison between groups was conducted using the nonparametric Mann-Whitney U test. * The chi-square test was used for the variable “smoker”.

**Table 3 ejihpe-15-00025-t003:** Bone health analysis using quantitative techniques according to copper dietary intake quartiles.

Variable	Copper Dietary Intake Quartile	n	Median	IQR	*p* Value
BUA (dB/MHz)	Q1 < 0.719	79	105.3	14.4	0.065
	Q2 (0.719–1.213)	78	104.8	15.6	
	Q3 > 1.213–3.403	78	106	11.5	
	Q4 > 3.403	77	102	13.7	
SOS (m/s)	Q1 < 0.719	79	1541.3	37.7	0.674
	Q2 (0.719–1.213)	78	1540	35.4	
	Q3 > 1.213–3.403	78	1543.8	35.7	
	Q4 > 3.403	77	1538.1	30.9	
pQCT Total Density (mg/cm^3^)	Q1 < 0.719	79	307.3	83.3	0.528
	Q2 (0.719–1.213)	78	295.6	69.7	
	Q3 > 1.213–3.403	78	298.5	77.7	
	Q4 > 3.403	78	297.8	56.6	
pQCT Trabecular Density (mg/cm^3^)	Q1 < 0.719	79	160.3	57.8	0.445
	Q2 (0.719–1.213)	78	146.7	60.9	
	Q3 > 1.213–3.403	78	161.2	51.3	
	Q4 > 3.403	78	166.1	42.2	
pQCT Cortical + Subcortical Density (mg/cm^3^)	Q1 < 0.719	79	431	133.6	0.171
	Q2 (0.719–1.213)	78	414.1	92.2	
	Q3 > 1.213–3.403	78	403	123.7	
	Q4 > 3.403	78	413.1	82.6	
pQCT Total Area (mm^2^)	Q1 < 0.719	79	299.7	51.4	0.929
	Q2 (0.719–1.213)	78	300.4	50.1	
	Q3 > 1.213–3.403	78	302.0	50.8	
	Q4 > 3.403	78	298.5	60.7	
pQCT Trabecular Area (mm^2^)	Q1 < 0.719	79	134.7	23.2	0.925
	Q2 (0.719–1.213)	78	135.1	22.7	
	Q3 > 1.213–3.403	78	135.8	23.0	
	Q4 > 3.403	78	134.2	27.2	
pQCT Cortical + Subcortical Area (mm^2^)	Q1 < 0.719	79	165	28.3	0.171
	Q2 (0.719–1.213)	78	165.3	27.3	
	Q3 > 1.213–3.403	78	166.7	27.8	
	Q4 > 3.403	78	164.4	33.7	
DXA Lumbar Spine (g/cm^2^)	Q1 < 0.719	79	0.888	0.227	0.866
	Q2 (0.719–1.213)	78	0.905	0.255	
	Q3 > 1.213–3.403	78	0.907	0.208	
	Q4 > 3.403	78	0.913	0.191	
DXA L2 (g/cm^2^)	Q1 < 0.719	79	0.865	0.256	0.944
	Q2 (0.719–1.213)	78	0.893	0.258	
	Q3 > 1.213–3.403	78	0.889	0.206	
	Q4 > 3.403	78	0.891	0.178	
DXA L3 (g/cm^2^)	Q1 < 0.719	79	0.885	0.269	0.821
	Q2 (0.719–1.213)	78	0.931	0.285	
	Q3 > 1.213–3.403	78	0.909	0.227	
	Q4 > 3.403	78	0.914	0.200	
DXA L4 (g/cm^2^)	Q1 < 0.719	79	0.892	0.203	0.762
	Q2 (0.719–1.213)	78	0.919	0.258	
	Q3 > 1.213–3.403	78	0.917	0.233	
	Q4 > 3.403	78	0.893	0.222	
DXA Femoral Neck (g/cm^2^)	Q1 < 0.719	79	0.750	0.171	0.369
	Q2 (0.719–1.213)	78	0.770	0.137	
	Q3 > 1.213–3.403	78	0.754	0.157	
	Q4 > 3.403	78	0.738	0.111	
DXA Femoral Trochanter (g/cm^2^)	Q1 < 0.719	79	0.609	0.144	0.427
	Q2 (0.719–1.213)	78	0.604	0.154	
	Q3 > 1.213–3.403	78	0.605	0.153	
	Q4 > 3.403	78	0.588	0.120	

Group comparisons were conducted using the nonparametric Kruskal-Wallis test.

**Table 4 ejihpe-15-00025-t004:** Multiple linear regression models.

Variable	Model 1		Model 2	
BUA (dB/MHz)	β (95% CI)	*p* Value	β (95% CI)	*p* Value
Q1 < 0.719	Reference		Reference	
Q2 (0.719–1.213)	−1.654 (−5.158; 1.849)	0.353	−0.899 (−4.149; 2.350)	
Q3 > 1.213–3.403	−2.481 (−5.984; 1.022)	0.164	−1.860 (−5.107; 1.388)	
Q4 > 3.403	−4.427 (−7.942; −0.912)	0.014	−1.851 (−5.185; 1.483)	
SOS (m/s)				
Q1 < 0.719	Reference			
Q2 (0.719–1.213)	−1.148 (−12.157; 9.861)	0.838		
Q3 > 1.213–3.403	1.076 (−9.933; 12.086)	0.848		
Q4 > 3.403	−2.713 (−13.758; 8.332)	0.629		
pQCT Total Density (mg/cm^3^)				
Q1 < 0.719	Reference			
Q2 (0.719–1.213)	(−8.44 (−26.14; 9.249)	0.348		
Q3 > 1.213–3.403	−7.681 (−25.38; 10.013)	0.394		
Q4 > 3.403	−13.431 (−31.13; 4.263)	0.136		
pQCT Trabecular Density (mg/cm^3^)				
Q1 < 0.719	Reference			
Q2 (0.719–1.213)	−3.991 (−17.023; 9.042)	0.547		
Q3 > 1.213–3.403	0.326 (−12.707; 13.358)	0.961		
Q4 > 3.403	4.250 (−8.782; 17.283)	0.522		
pQCT Cortical + Subcortical Density (mg/cm^3^)				
Q1 < 0.719	Reference		Reference	
Q2 (0.719–1.213)	−10.26 (−36.42; 15.902)	0.441	−4.154 (−28.253; 19.944)	0.735
Q3 > 1.213–3.403	−13.16 (−39.33; 12.997)	0.323	−8.582 (−32.662; 15.498)	0.484
Q4 > 3.403	−28.79 (−54.95; −2.626)	0.031	−10.412 (−35.059; 14.235)	0.406
pQCT Total Area (mm^2^)				
Q1 < 0.719	Reference			
Q2 (0.719–1.213)	−4.203 (−19.66; 11.253)	0.593		
Q3 > 1.213–3.403	−2.209 (−17.66; 13.246)	0.779		
Q4 > 3.403	−7.297 (−22.75; 8.158)	0.354		
pQCT Trabecular Area (mm^2^)				
Q1 < 0.719	Reference			
Q2 (0.719–1.213)	−1.751 (−8.684; 5.182)	0.620		
Q3 > 1.213–3.403	−0.978 (−7.911; 5.955)	0.782		
Q4 > 3.403	−3.45 (−10.383; 3.484)	0.328		
pQCT Cortical + Subcortical Area (mm^2^)				
Q1 < 0.719	Reference			
Q2 (0.719–1.213)	−2.424 (−10.973; 6.125)	0.577		
Q3 > 1.213–3.403	−1.404 (−9.954; 7.145)	0.747		
Q4 > 3.403	−4.484 (−1.033; 4.065)	0.303		
DXA Lumbar Spine (g/cm^2^)				
Q1 < 0.719	Reference			
Q2 (0.719–1.213)	0.015 (−0.040; 0.070)	0.593		
Q3 > 1.213–3.403	−0.013 (−0.069; 0.042)	0.630		
Q4 > 3.403	−0.024 (−0.079; 0.031)	0.393		
DXA Hip (g/cm^2^)				
Q1 < 0.719	Reference			
Q2 (0.719–1.213)	0.012 (−0.026; 0.050)	0.527		
Q3 > 1.213–3.403	−0.005 (−0.043; 0.033)	0.798		
Q4 > 3.403	−0.026 (−0.064; 0.012)	0.180		

Model 1 (unadjusted). Model 2 adjusted for age and weight.

## Data Availability

The dataset analyzed in the current study is not publicly available due to national data regulations and for ethical reasons, including that we do not have the explicit written consent of the study volunteers to make their deidentified data available at the end of the study. However, datasets and SPSS statistical analyses can be requested by sending a letter to the corresponding author.

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
