# Peer review of "The Role of Copper Intake in Bone Health: A Quantitative Analysis in Postmenopausal Spanish Women"

_ejihpe, 2025, doi:10.3390/ejihpe15020025_

Round 1

Reviewer 1 Report

Comments and Suggestions for Authors

The manuscript deals with the association between dietary copper intake and bone health in postmenopausal women through densitometric assays. The authors highlighted as a point of strength the use of three distinct quantitative techniques for assessing bone health in postmenopausal Spanish women determining bone mineral density. The study does not reveal an association between copper intake and bone health in 313 postmenopausal Spanish women. This conclusion counteracts those reported for similar observational studies in which a reduced risk of osteoporosis is related to higher copper intake. The limited number of analised samples in the work can explain the different results as hypothesized by the same authors of the work. This point of weakness is crucial, it  suggests that the work cannot be recommended for publication

Author Response

Comments 1: The manuscript deals with the association between dietary copper intake and bone health in postmenopausal women through densitometric assays. The authors highlighted as a point of strength the use of three distinct quantitative techniques for assessing bone health in postmenopausal Spanish women determining bone mineral density. The study does not reveal an association between copper intake and bone health in 313 postmenopausal Spanish women. This conclusion counteracts those reported for similar observational studies in which a reduced risk of osteoporosis is related to higher copper intake. The limited number of analised samples in the work can explain the different results as hypothesized by the same authors of the work. This point of weakness is crucial, it  suggests that the work cannot be recommended for publication

Response 1: The present study emphasizes the significance of reporting negative results. Although the manuscript reveals no correlation between dietary copper intake and bone quality in 313 postmenopausal Spanish women, the manuscript enhances the database knowledge of the wider scientific field. Negative findings are critical for evidence-based practice since they are significantly helpful in hypothesis refinement, research agenda definition and identification of needs for large-scale replication of studies. In the discussion section of the manuscript, we explicitly state its limitations, mainly the small sample size. This transparency improves our credibility and also is good scientific practice. Reporting investigations with null results extends from outstanding importance for several reasons. First, they reduce the influence of publication bias, under which the literature base only contains positive results on the effects and significance of particular interventions or relations. Second, negative results tend to question existing assumptions, stimulate question-asking, and provide motivation to look for additional explanations or procedures. Third, such studies provide a more comprehensive view of variability as an inevitable part of research, and make sure that any conclusions derived from the evidence are truly as strong and confident as possible. Thus, the application of three different quantitative indices of bone health from a methodological point of view strengthens the relevance of the work, even if there are no statistically significant results. As such, rather than detracting from the current field, our study builds upon it by highlighting the need for further studies as well as more elaborate datasets to clarify these associations.

Reviewer 2 Report

Comments and Suggestions for Authors

Dear Authors,

It is an honor to review this scientific article, which presents valuable insights into the relationship between copper intake and bone health in postmenopausal women.

The study's comprehensive approach, using multiple quantitative techniques, provides a dynamic perspective on this important topic.

I appreciate the opportunity to contribute to the understanding of this research, and I look forward to its potential impact on the field of bone health.

Please:

1. Introduction

Lines 55-75 (Please, see APA Style 7th):

“Copper is a trace element of vital importance for sustaining human health (I. 55 Scheiber, Dringen, & Mercer, 2013). Copper is necessary to maintain healthy growth and 56 development and helps nourish bones, the brain, cardiovascular function and other vital 57 organs. The human body cannot make copper, and we must rely entirely on the diet for 58 this essential trace mineral (Cui, Yan, et al., 2024; Fan, Ni, & Zhang, 2022; Li et al., 2023; I F. Scheiber, Mercer, & Dringen, 2014; Turnlund, 1998). There have been few observational 60 studies regarding the association between copper and the osteoporosis risk, as previous 61 studies have largely examined the association between serum copper and osteoporosis 62 (Chaudhri, Kemmler, Harsch, & Watling, 2009; Mahdavi-Roshan, Ebrahimi, & Ebrahimi, 63 2015; Qu et al., 2018). However, recent studies have investigated the relationship between 64 dietary copper intake and bone health in humans and attempted to determine how copper 65 may be involved in safeguarding bone integrity and reducing osteoporosis risk (M. Chen, 66 Jia, & Gao, 2024; Cui, Yan, et al., 2024; Fan et al., 2022; Pasco et al., 2024). Copper is an 67 important constituent of several physiological processes, the most important being the 68 incorporation of copper into enzymes essential for energy metabolism and the formation 69 of connective tissue crosslinks, particularly in bone (Rył et al., 2021). Copper supports os-70 teogenesis by promoting the differentiation of bone mesenchymal stem cells towards bone 71 formation rather than adipogenesis (Rodríguez, Ríos, & González, 2002). In addition, cop-72 per deficiency can result in Menkes' disease, with osteoporosis being one of its primary 73 adverse effects (J. Chen et al., 2020M. Chen et al., 2024; Panichsillaphakit, Kwanbun-74 bumpen, Chomtho, & Visuthranukul, 2022).”

Background:

It is important to consider a more robust “state of the art” when conducting research.  

A comprehensive review of existing literature ensures that new studies are built on solid foundations, identifying gaps, validating methodologies, and providing context for interpreting results.

So, given the above:

What are some of the health areas where copper intake is being researched, aside from nail growth and how was the copper intake of participants measured in the study?

What were the findings in the unadjusted regression analysis regarding copper intake and bone health and why did the associations between copper intake and bone health lose significance after adjusting for age and weight?

What is the gap in research highlighted by the authors regarding dietary copper intake and osteoporosis or low bone mass?

2.2. Heel Quantitative Ultrasound (QUS) Assessment. ( Line 94):

Quantitative ultrasound (QUS) was conducted using the Sahara Clinical Sonometer 95 (Hologic, Bedford, Massachusetts) following a standardized protocol. Trained staff en- 96sured that the participants could complete measurements on both heels, excluding those 97with open wounds, injuries, or metal implants in the heel. Daily quality control was per- 98 formed using a phantom, as per the manufacturer’s guidelines. The device measured the 99 speed of sound (SOS) and broadband ultrasound attenuation (BUA), indicators of bone 100 health, with higher values indicating better bone quality (Please, insert studies/references)

2.3. Dual-Energy X-ray Absorptiometry (DXA) Assessment (line 102)

BMD at the femoral neck (FN), femoral trochanter (FT) and L2–L4, as well as at a 103combined L2, L3, and L4 region, was measured via dual-energy X-ray absorptiometry 104(DXA). Body weight and body height were also recorded, and body mass index (BMI) was 105 also calculated. BMD values were assessed via densitometry with a NORLAND XR-800 106 device (Norland Medical Systems Inc.). The BMD values are expressed in grams per 107 square centimetre. 108 (Please, insert studies/references)

2.4. Peripheral Quantitative Computed Tomography (pQCT) Assessment (line 109)

Peripheral quantitative computed tomography (pQCT) scans of the nondominant 110 distal forearm were obtained using a Stratec XCT-2000 scanner (Stratec Medizintechnik, 111 Pforzheim, Germany). At 4% of the total forearm length, an image was taken with the 112 scanner positioned at the distal end of the forearm. The data from the XCT-2000 scans 113 were processed using the software package (version 5.50) provided by the manufacturer. 114 The pQCT scans provide a volumetric measurement of bone mineral density and allow 115 differentiation between trabecular and cortical bone (Please, insert studies/references)

2.6. Statistical Analysis.

Please, insert IBM SPSS Statistics

4. Discussion

It is important to consider a more robust discussion of the results. It allows for a deeper understanding of the results, offering insights into their significance and how they contribute to advancing knowledge in the field.

So, given the above:

How does this study differ from previous research and what have previous studies found regarding the relationship between serum copper levels and bone health, particularly in men?

Why is further research needed to clarify the association between copper intake and bone health, according to the authors?

How does the use of different methodologies to assess bone health enhance the comprehensiveness of the study?

How might the small sample size in the study affect the results?

What might be the next steps for future research based on the limitations and strengths of this study?

Thanks! 

Kind regards

Author Response

Comment 1: 1. Introduction

Lines 55-75 (Please, see APA Style 7th):

“Copper is a trace element of vital importance for sustaining human health (I. 55 Scheiber, Dringen, & Mercer, 2013). Copper is necessary to maintain healthy growth and 56 development and helps nourish bones, the brain, cardiovascular function and other vital 57 organs. The human body cannot make copper, and we must rely entirely on the diet for 58 this essential trace mineral (Cui, Yan, et al., 2024; Fan, Ni, & Zhang, 2022; Li et al., 2023; I F. Scheiber, Mercer, & Dringen, 2014; Turnlund, 1998). There have been few observational 60 studies regarding the association between copper and the osteoporosis risk, as previous 61 studies have largely examined the association between serum copper and osteoporosis 62 (Chaudhri, Kemmler, Harsch, & Watling, 2009; Mahdavi-Roshan, Ebrahimi, & Ebrahimi, 63 2015; Qu et al., 2018). However, recent studies have investigated the relationship between 64 dietary copper intake and bone health in humans and attempted to determine how copper 65 may be involved in safeguarding bone integrity and reducing osteoporosis risk (M. Chen, 66 Jia, & Gao, 2024; Cui, Yan, et al., 2024; Fan et al., 2022; Pasco et al., 2024). Copper is an 67 important constituent of several physiological processes, the most important being the 68 incorporation of copper into enzymes essential for energy metabolism and the formation 69 of connective tissue crosslinks, particularly in bone (Rył et al., 2021). Copper supports os-70 teogenesis by promoting the differentiation of bone mesenchymal stem cells towards bone 71 formation rather than adipogenesis (Rodríguez, Ríos, & González, 2002). In addition, cop-72 per deficiency can result in Menkes' disease, with osteoporosis being one of its primary 73 adverse effects (J. Chen et al., 2020; M. Chen et al., 2024; Panichsillaphakit, Kwanbun-74 bumpen, Chomtho, & Visuthranukul, 2022).”

Response 1: 

We sincerely apologize for the oversight. Zotero was used as our reference management tool, and we were confident that the formatting adhered to APA 7th guidelines. The errors identified in the introduction section have now been corrected.

Comment 2: Background:

It is important to consider a more robust “state of the art” when conducting research.  

A comprehensive review of existing literature ensures that new studies are built on solid foundations, identifying gaps, validating methodologies, and providing context for interpreting results.

So, given the above:

What are some of the health areas where copper intake is being researched, aside from nail growth and how was the copper intake of participants measured in the study?

What were the findings in the unadjusted regression analysis regarding copper intake and bone health and why did the associations between copper intake and bone health lose significance after adjusting for age and weight?

What is the gap in research highlighted by the authors regarding dietary copper intake and osteoporosis or low bone mass?

Response 2: We sincerely thank the reviewer for their insightful feedback and valuable suggestions. In response to Comment 2, we have expanded the Introduction section to address all the issues raised. The revised section now provides a more comprehensive context and aligns closely with the reviewer's recommendations. We greatly appreciate your guidance in enhancing the quality of our manuscript.

Comment 3: 2.2. Heel Quantitative Ultrasound (QUS) Assessment. ( Line 94):

Quantitative ultrasound (QUS) was conducted using the Sahara Clinical Sonometer 95 (Hologic, Bedford, Massachusetts) following a standardized protocol. Trained staff en- 96sured that the participants could complete measurements on both heels, excluding those 97with open wounds, injuries, or metal implants in the heel. Daily quality control was per- 98 formed using a phantom, as per the manufacturer’s guidelines. The device measured the 99 speed of sound (SOS) and broadband ultrasound attenuation (BUA), indicators of bone 100 health, with higher values indicating better bone quality (Please, insert studies/references)

Response 3: Done. 

Comment 4: 2.3. Dual-Energy X-ray Absorptiometry (DXA) Assessment (line 102)

BMD at the femoral neck (FN), femoral trochanter (FT) and L2–L4, as well as at a 103combined L2, L3, and L4 region, was measured via dual-energy X-ray absorptiometry 104(DXA). Body weight and body height were also recorded, and body mass index (BMI) was 105 also calculated. BMD values were assessed via densitometry with a NORLAND XR-800 106 device (Norland Medical Systems Inc.). The BMD values are expressed in grams per 107 square centimetre. 108 (Please, insert studies/references)

Response 4: Done

Comment 5: .4. Peripheral Quantitative Computed Tomography (pQCT) Assessment (line 109)

Peripheral quantitative computed tomography (pQCT) scans of the nondominant 110 distal forearm were obtained using a Stratec XCT-2000 scanner (Stratec Medizintechnik, 111 Pforzheim, Germany). At 4% of the total forearm length, an image was taken with the 112 scanner positioned at the distal end of the forearm. The data from the XCT-2000 scans 113 were processed using the software package (version 5.50) provided by the manufacturer. 114 The pQCT scans provide a volumetric measurement of bone mineral density and allow 115 differentiation between trabecular and cortical bone (Please, insert studies/references)

Response 5: Done

Comment 6: 2.6. Statistical Analysis.

Please, insert IBM SPSS Statistics

Response 6: We are unsure why we need to include a reference to IBM SPSS, as we have used JASP for our analysis.

Comment 7: 4. Discussion

It is important to consider a more robust discussion of the results. It allows for a deeper understanding of the results, offering insights into their significance and how they contribute to advancing knowledge in the field.

So, given the above:

How does this study differ from previous research and what have previous studies found regarding the relationship between serum copper levels and bone health, particularly in men?

Response 7: In the present study, we did not analyze serum copper levels, and therefore, our research does not directly contribute to this specific area of investigation. While we have included a discussion of the existing literature on serum copper levels to provide context for our findings, it is important to clarify that we do not have data regarding serum copper concentrations in the patients included in our study. 

Comment 8: Why is further research needed to clarify the association between copper intake and bone health, according to the authors?

Response 8: We have expanded the discussion to emphasize the need for further research, including longitudinal studies and meta-analyses, to address conflicting findings and clarify the role of copper intake in bone health.

Comment 9: How does the use of different methodologies to assess bone health enhance the comprehensiveness of the study?

Response 9: We have expanded the discussion to highlight how using QUS, DXA, and pQCT enhances the study's comprehensiveness by providing a multidimensional assessment of bone health.

Comment 10: How might the small sample size in the study affect the results?

Response 10: We acknowledge the issue of small sample size as a clear limitation of our study, as stated in the limitations section. This is associated with an increased risk of a Type II error, potentially reducing the power to detect significant associations.

Comment 11: What might be the next steps for future research based on the limitations and strengths of this study?

Response 11: We believe that the next steps for future research have already been addressed in the discussion section in response to previous comments.

Round 2

Reviewer 1 Report

Comments and Suggestions for Authors

The reply of authors does not allow me to change my suggestions

Author Response

We are very grateful for your response.

Reviewer 2 Report

Comments and Suggestions for Authors

Dear Authors,

Thank you very much for responding to my request so promptly!

Kind regards

Author Response

We are very grateful for your response.